# Hepatocyte-Specific Triggering of Hepatic Stellate Cell Profibrotic Activation by Apoptotic Bodies: The Role of Hepatoma-Derived Growth Factor, HIV, and Ethanol

**DOI:** 10.3390/ijms24065346

**Published:** 2023-03-10

**Authors:** Moses New-Aaron, Siva Sankar Koganti, Murali Ganesan, Sharma Kanika, Vikas Kumar, Weimin Wang, Edward Makarov, Kusum K. Kharbanda, Larisa Y. Poluektova, Natalia A. Osna

**Affiliations:** 1Department of Environmental Health, Occupational Health and Toxicology, College of Public Health, University of Nebraska Medical Center, Omaha, NE 68198, USA; 2Research Service, Veterans Affairs Nebraska-Western Iowa Health Care System, Omaha, NE 68105, USA; 3Department of Internal Medicine, University of Nebraska Medical Center, Omaha, NE 68105, USA; 4Department of Genetics Cell Biology & Anatomy, University of Nebraska Medical Center, Omaha, NE 68105, USA; 5Department of Pharmacology and Experimental Neuroscience, University of Nebraska Medical Center, Omaha, NE 68105, USA

**Keywords:** apoptotic bodies, HIV, alcohol, hepatic stellate cells, HDGF, liver fibrosis

## Abstract

Liver disease is one of the leading comorbidities in HIV infection. The risk of liver fibrosis development is potentiated by alcohol abuse. In our previous studies, we reported that hepatocytes exposed to HIV and acetaldehyde undergo significant apoptosis, and the engulfment of apoptotic bodies (ABs) by hepatic stellate cells (HSC) potentiates their pro-fibrotic activation. However, in addition to hepatocytes, under the same conditions, ABs can be generated from liver-infiltrating immune cells. The goal of this study is to explore whether lymphocyte-derived ABs trigger HSC profibrotic activation as strongly as hepatocyte-derived ABs. ABs were generated from Huh7.5-CYP2E1 (RLW) cells and Jurkat cells treated with HIV+acetaldehyde and co-culture with HSC to induce their pro-fibrotic activation. ABs cargo was analyzed by proteomics. ABs generated from RLW, but not from Jurkat cells activated fibrogenic genes in HSC. This was driven by the expression of hepatocyte-specific proteins in ABs cargo. One of these proteins is Hepatocyte-Derived Growth Factor, for which suppression attenuates pro-fibrotic activation of HSC. In mice humanized with only immune cells but not human hepatocytes, infected with HIV and fed ethanol, liver fibrosis was not observed. We conclude that HIV+ABs of hepatocyte origin promote HSC activation, which potentially may lead to liver fibrosis progression.

## 1. Introduction

When many thought Human Immuno-deficiency Virus/Acquired Immune Deficiency Syndrome (HIV/AIDS) was a relic of the past, the incidence of HIV-related multi-organ dysfunction began to emerge among people living with HIV (PLWH) [1,2]. This is due to antiretroviral therapy (ART)-induced longevity, which conversely allows the incidence of age-related comorbidities not previously observed among PLWH. Given the impaired quality of life and outrageous incremental care cost for PLWH due to non-infection comorbidities [2,3,4,5], such as cardiovascular disease, liver disease, and alcohol use disorder [6,7,8], the management of PLWH in the ART era has become a critical economy and public health issue [5]. In response to this dilemma, our group explored the pathogenicity of HIV-induced liver injury in alcohol settings. We demonstrated the highest hepatocyte apoptosis induced by the combined exposure to acetaldehyde and HIV [9,10], which exceeds the effects of single exposures. The apoptotic bodies (ABs) derived from these cells express enormous HIV proteins, HIV RNA, and malondialdehyde (MDA), capable of inducing hepatic stellate cell (HSC) profibrotic activation after ABs internalization [11]. While HSC profibrotic activation and the consequent extracellular matrix (ECM) deposition are homeostatic responses to hepatocyte depletion [12], acetaldehyde- and HIV-induced massive hepatocyte depletion engenders excessive continuous ECM deposition, which is a pivotal event in the incidence of hepatic fibrosis [13]. Thus, the crosstalk between hepatocytes and HSC provides an important axis for alcohol and HIV-induced liver fibrosis development [14]. 

A brief review of this crosstalk must be established as we commence hepatocyte apoptosis. Unlike other studies, which utilized the UV method to generate hepatocyte ABs [15,16], in the current study, we generated ABs from acetaldehyde and HIV-treated hepatocyte-like cells (RLW cells) [11]. While we did not seek to discredit the UV method of ABs generation, we preferred to mimic natural exposures for ABs generation to avoid the possibility that in HIV-infected hepatocytes, UV may affect ABs cargo [17]. Moreover, our mode of ABS generation better reproduces the natural generation of hepatocyte ABs in alcohol-abusing PLWH.

As we demonstrated before, there is an internalization of the phosphatidylserine-expressing hepatocyte-derived ABs by AXL, a phosphatidylserine recognition receptor expressed on HSC leading to their profibrotic activation [11]. However, a profibrotic activation may not be unique to the liver and require the involvement of hepatic ABs. In fact, activation of fibroblasts in various organs has been observed and is mediated by the internalization of epithelial or endothelial-derived ABs [18,19]. In addition, internalization of ABs derived from immune cells has been reported [20]. As we established, in the liver, ABs from hepatocytes serve as a non-immune source for HSC profibrotic activation [9,11]. However, the liver is a graveyard for activated immune cells, which accumulate many HIV-infected circulating lymphocytes coming to the liver to die [21,22]. It is unknown whether in alcohol-abusing PLWH, profibrotic activation of HSC is induced by the engulfment of any HIV-expressing ABs or whether these ABs should be strictly hepatocyte specific. Hence, in this study, we explore whether lymphocyte-derived ABs trigger HSC profibrotic activation as strongly as hepatocyte-derived ABs. This knowledge will help to identify the role of HIV-infected hepatocytes and the involved cargo components for ameliorating alcohol- and HIV-induced liver injury. 

## 2. Results

### 2.1. HSC Profibrotic Activation after Internalization of Acetaldehyde- and HIV-Derived Hepatocyte ABSS

We previously observed robust hepatocyte (RLW) apoptosis from the combination of the acetaldehyde-generating system (AGS) and HIV [9], followed by the generation of HIV and MDA-expressing ABs (AB_AGS+HIV(RLW)_) to induce profibrotic activation in HSC (LX2 cells) [11]. Here, LX2 cells were exposed to hepatic AB_AGS+HIV(RLW)._ The design of these in vitro experiments as well as cell characterization and the details on AGS are described in Materials and Methods. 

First, we found that exposure of HIV-infected RLW cells to AGS provided a six-fold induction of HIV gag RNA expression in AB-captured LX2 cells compared LX2 cells exposed to ABs derived from HIV-infected but AGS non-exposed RLW cells (AB_HIV(RLW)_), as shown in Figure 1a. Moreover, internalization of AB_AGS+HIV(RLW)_ increased *COL1A1* mRNA by two-fold in LX2 cells normalized to ABs from untreated RLW cells (AB_CONTROL(RLW)_); see Figure 1b. Similarly, internalization of AB_AGS+HIV(RLW)_ increased the Tissue Inhibitor of Metalloproteinases (*TIMP*)*-1* mRNA in LX2 cells by two-fold (Figure 1c) and three-fold in Transformation Growth Factor Beta (*TGFβ*) (Figure 1d).

### 2.2. Lack of Profibrotic Changes in HSC after Internalization of ABs Derived from HIV- and AGS-Exposed Lymphocytes 

In a physiological system, HSC encounters lymphocyte ABs as frequently as hepatocyte ABS. Therefore, lymphocyte ABs were generated from lymphocyte-like cells (Jurkat cells) treated with AGS and HIV to yield ABs (AB_AGS+HIVJK_). For these studies, we used the same number of AB_AGS+HIVJK_ and the same ABs: HSC ratio as RLW-derived ABs. This was to determine if lymphocyte ABs provided similar profibrotic effects as hepatocyte ABs after engulfment by HSC. To accomplish this, ABs_AGS+HIVJK_ were introduced to LX2 cells, and HIV gag RNA was quantified. In agreement with the effects provided by AB_AGS+HIV(RLW)_, AB_AGS+HIVJK_ increased HIV gag RNA by two-fold when compared to LX2 cells that internalized ABs derived from HIV-infected Jurkat cells in the absence of AGS (ABS_HJK_); see Figure 2a. However, there were no significant profibrotic changes in *COL1A1* mRNA, Matrix Metalloproteinases (*MMP*)*-2* mRNA, and *TGFβ* mRNAs in LX2 cells after the internalization of AB_AGS+HIVJK_ as compared to LX2 cells exposed to ABs from untreated Jurkat cells (ABS_CJK_); see Figure 2b–d. These findings were opposite from what was observed when ABs from the treated RLW cells were used. 

### 2.3. Proteins Expressed in ABs_AGS+HIV(RLW)_ and ABs_AGS+HIV(JK)_ Associated with Hepatotoxicity 

A proteomic analysis was performed to compare the proteins in lymphocyte ABs with hepatocyte ABs. We used ABs generated from each cell type under exposure to AGS+HIV since this treatment demonstrated the highest apoptosis and the highest profibrotic effects in LX2 cells after ABs engulfment. A total of 2000 proteins were fully identified in lymphocyte and hepatocyte ABs. Principal component analysis for all the profiled proteins showed that the AB_AGS+HIV(RLW)_ and ABS_AGS+HIVJK_ groups clustered separately from each other, suggesting that the protein cargo in AB_AGS+HIV(RLW)_ vs. AB_AGS+HIVJK_ is different (Figure 3a). In fact, 600 proteins of AB_AGS+HIV(RLW)_ were significantly different from proteins in AB_AGS+HIVJK_. Furthermore, the ingenuity pathway analysis (IPA) unveiled 47 proteins of AB_AGS+HIV(RLW)_ vs. AB_AGS+HIVJK,_ which were associated with liver fibrosis (Figure 3b). 

### 2.4. Proteins in Hepatocyte ABS vs. Lymphocyte ABs Associated with Liver Fibrosis

Further analysis of the proteins in lymphocyte ABs vs. hepatocyte ABs through the IPA showed that 44% of proteins associated with liver fibrosis were significantly upregulated in AB_AGS+HIV(RLW)_ vs. AB_AGS+HIVJK_; see Figure 4a_._ Given that ABs were exogenously introduced to LX2 cells in our experimental design, only proteins capable of triggering HSC profibrotic activation after an exogenous application are relevant to this study. Numerous proteins were differentially expressed in ABs from both cell types. In this study, we chose to concentrate on Hepatoma-Derived Growth Factor (HDGF), since as shown by others, HDGF has been identified as a protein capable of triggering HSC profibrotic activation after exogenous application [23]. HDCF was upregulated in AB_AGS+HIV(RLW)_ vs. AB_AGS+HIVJK_; see Figure 4b. 

Given that hepatocyte ABs used in this study were generated from RLW cells, which is the hepatoma cell line, for translational interpretation, we needed to confirm the presence of HDGF in primary human hepatocytes (PHH). While HDGF was expressed in PHH, its amount in RLW cells was 50% higher. However, as indicated by Figure 4b, AB_AGS+HIVJK_, do not express HDGF, and thus, the finding was not related to the hepatoma origin of RLW cells. 

### 2.5. Suppression of Profibrotic Genes in LX2 Cells after Internalization of ABs_AGS+HIV_ Derived from HDGF-Knockdown RLW Cells

To investigate the input of HDGF as a trigger for LX2 profibrotic activation after the internalization of AB_AGS+HIV(RLW)_, HDGF knockdown was performed by HDGF siRNA transfection. RLW cells were cultured with either HDGF siRNA or control siRNA as a negative control. Approximately, a 25% reduction in HDGF protein expression was observed in HDGF siRNA-transfected RLW cells compared to those transfected with the control siRNA (Figure 5a,b). 

To examine the role of HDGF in LX2 profibrotic activation after the internalization of ABS_AGS+HIV(RLW)_, HDGF knockdown in RLW cells was performed as described for RLW cells. After HDGF knockdown, RLW cells were treated with HIV and AGS to generate ABs. Then, profibrotic genes were evaluated in LX2 cells exposed to AB_AGS+HIV_ from HDGF knockdown RLW cells (HDGF siRNA_ABAGS+HIV_). As a negative control, LX2 cells were exposed to AB_AGS+HIV_ derived from RLW cells transfected with control siRNA (control siRNA_ABAGS+HIV_). Two-fold suppression of *COL1A1* and *TIMP1* mRNA in LX2 cells was observed after the internalization of HDGF siRNA_ABAGS+HIV_ compared to LX2 cells that internalized control siRNA_ABAGS+HIV_ (Figure 6a,b). This was also confirmed by a 50% suppression of *TGFβ* mRNA in LX2 cells after the engulfment of HDGF siRNA_ABAGS+HIV_ (Figure 6c). On the opposite, additional analysis showed that *MMP2* mRNA was approximately four times higher in LX2 cells after the internalization of HDGF siRNA_ABAGS+HIV_ (Figure 6d). 

### 2.6. Alcohol Consumption Exacerbated HIV-1-Induced Immunopathology but Did Not Affect Liver Fibrosis in the Absence of Transplanted Human Hepatocytes

To completely exclude lymphocyte ABs as a trigger for HSC profibrotic activation, we used an in vivo study. NSG mice were fed ethanol in drinking water as the only available fluid, starting at 5% weight/volume (*w*/*v*), and subsequently, were gradually increased to 10%, 15%, and 20% *w*/*v*. Mice were then maintained at 20% *w*/*v*, a level that models chronic alcohol use in humans. The control and “alcoholic” mice were infected with HIV-1_ADA_. Alcohol consumption at 20% *w*/*v* over 20 days reduced the weight of animals (Figure 7A). We observed a 10-times higher HIV-1 viral load in peripheral blood and a significant reduction in the CD4+ cells compared to baseline numbers (Figure 7B), which is a mark of HIV-1-induced immunopathology. While HIV-1-infected animals on regular water did not show a significant reduction in CD4+ cells by 5 weeks of infection, EtOH decreased the CD4+ cells’ number by 5 weeks of HIV infection (Figure 7D–F). Nevertheless, we did not observe changes in ALT and AST activity due to alcohol consumption (Figure 7C).

### 2.7. Alcohol Consumption Increased Human Immune Cells Infiltration but Did Not Induce Fibrotic Changes in the Liver of HIV-1-Infected Humanized Mice

We previously demonstrated the development of hepatic fibrosis in immunodeficient NSG mice injected with HIV-containing RLW-derived ABs [11]. However, nothing is known about the in vivo effects of lymphocyte ABs on the liver. Therefore, we evaluated liver pathology for the effects of lymphocyte ABs by staining NSG liver tissue collected at 5 weeks post-infection and alcohol consumption at 20% *w*/*v*. We used anti-HLA-DR antibodies to detect human immune cells, and infected cells were detected by staining for HIVp24 antigen. We found very few human immune cells in the control animals. On the contrary, alcohol triggered significant liver infiltration by human cells, and a significant proportion of them was HIV-1 infected. Some cells appeared as apoptotic. Nevertheless, we did not find signs of profibrotic changes by staining for activated stellate cells with a-SMA, and collagen deposition was not observed by staining with Sirius Red (Figure 8). 

## 3. Discussion

Development of liver fibrosis is a significant clinically-relevant problem in HIV infection, and the risk of progression to end-stage liver disease in PLWH is more prominent in alcohol abusers [24]. As revealed from our previous studies, the combined treatment of RLW cells with AGS and HIV induced the highest rates of hepatic apoptotic cell death, and these data were confirmed on PHH exposed to HIV and ethanol [9]. This allowed us to use cultured hepatocyte-like RLW cells for the generation of ABs instead of PHH since the supply of human primary cells is very limited and they undergo fast de-differentiation, thus losing expression of ethanol-metabolizing enzymes. This contributes to liver fibrosis development, as shown in various models including alcoholic hepatitis, hepatocyte toxicity, and namely, hepatocyte apoptosis [25]. 

The liver attracts activated immune cells which undergo intensive apoptosis in this organ [21,26]. Consequently, HIV RNAs/HIV proteins released from these lymphocytes can non-productively infect hepatocytes, and their intracellular accumulation induced by exposure to ethanol metabolites further promotes hepatotoxicity [9,10]. Ethanol induces oxidative stress and triggers the rise of HIV intracellular levels either via the enhancement of HIV replication in lymphocytes or due to the stabilization of HIV proteins because of the suppression of their degradation in hepatocytes [10,27], ultimately leading to apoptotic cell death. The question we asked in this study was whether the fibrogenic activation of HSC by engulfment of ABSs is predominantly hepatocyte specific. In other words, are there any differences in HSC activation by ABs generated from hepatocytes and lymphocytes? To clarify this point, we used previously identified conditions that provide the highest apoptosis induction and ABs generation in hepatocytes, which are the co-treatment of HIV-infected cells with AGS [14]. For consistency, Jurkat cells were treated in a similar way with AGS and HIV. Then, ABs from both sources were applied in the same quantities to HSC to induce profibrotic activation. Surprisingly, while ABs from hepatocytes activated pro-fibrotic genes in HSC, ABs from Jurkat cells showed no such effects. Similar results were observed when ABs were generated from HIV-infected human primary lymphocytes [9].

In our previous studies, we reported that the fibrogenic activation of HSC was more potent when hepatocyte-derived ABs expressed HIV RNAs/proteins and an oxidative stress marker, malondialdehyde (MDA) [9,14]. However, here, Jurkat cells also expressed HIV proteins. Furthermore, HIV gag protein expression by LX2 (HSC) cells after engulfment of ABs generated from either HIV vs. HIV+AGS-treated Jurkat cells or RLW cells was higher upon exposure to AGS. This indicates that in addition to HIV proteins as possible triggers of HSC activation, there are some hepatocyte-specific proteins that contribute to the induction of pro-fibrotic genes in HSC.

By mass spectrometry, we were able to identify about 600 proteins differentially expressed in HIV-infected RLW and Jurkat cells, both treated with AGS. Among these proteins was HDGF, which is more potently expressed in malignant cells, but non-malignant cells (hepatocytes) also contain this protein, as evident from the publications and our data [28,29]. HDGF is related to liver fibrosis [29,30]. It was identified as a growth factor for hepatocytes and some other cancer cells and activates the signaling via the ERK 1/2 and STAT3 pathways [31,32]. As we reported before, the pro-fibrotic activation of HSC by engulfment of ABs derived from HIV+AGS-treated hepatocytes occurs via these pathways [11]. Here, by knocking down HDGF with siRNA transfection, we suppressed *COL1A1*, *TGFβ*, and *TIMP1* as well as increased *MMP2* gene activation, thereby attenuating fibrogenic effects in HSC. Thus, we believe that HDGF may serve as one of the pro-fibrotic hepatocyte-specific signals for HSC activated by ABs internalization. Based on mass spectrometry data, this protein is not expressed in Jurkat cells. There is no doubt that HDGF is not the only protein expressed in ABs for HSC activation. However, although in this study, we validated only this protein, there are other ABs cargo components that are responsible for hepatocyte-specific liver fibrosis development under exposure to HIV and alcohol. 

The fact that AB-dependent activation of HSC leads to the progression of liver fibrosis was confirmed by our in vivo mouse study where the highest levels of liver damage, pro-fibrogenic gene induction, and Sirius Red positive staining were observed in liver tissue from ethanol-fed mice injected with human HIV+ hepatocyte-derived ABs [11]. On the contrary, in the in vivo study shown here, mice humanized only with the immune system, HIV-infected, and fed an ethanol diet demonstrated no liver fibrosis in the absence of transplanted human hepatocytes. This is not because this mode of ethanol in water feeding failed to achieve liver damage and initial steps of fibrosis development [33,34]. However, mouse hepatocytes cannot be infected with human HIV and thus, do not undergo significant apoptosis under ethanol exposure. In this case, we assume that HSC were not activated by the engulfment of HIV+ hepatocyte ABs which contain hepatocyte-specific proteins (such as HDGF) sensed by HSC, and liver fibrosis is not activated. 

We cannot exclude that in addition to the role of crosstalk between hepatocytes and HSC in liver fibrosis development, it is further potentiated by the interactions between HSC and liver macrophages as has been demonstrated by others on the model of HIV-HCV co-infection [35]. In our hands, the internalization of HIV+ hepatocyte ABs by macrophages leads to inflammasome induction in these cells [9], which in turn, contributes to liver fibrosis activation [36,37]. However, in HIV infection, exposure of hepatocytes to HIV followed by AB formation is crucial for alcohol-induced progression to liver fibrosis as evident from the results of both in vitro and in vivo studies.

## 4. Materials and Methods

### 4.1. Reagents and Antibodies

High-glucose Dulbecco’s Modified Eagle Medium (DMEM) and fetal bovine serum (FBS) were purchased from Invitrogen (Waltham, MA, USA); Trizol was purchased from Life Technologies (Carlsbad, CA, USA); primer probes, high-capacity reverse transcription kit, and real-time polymerase chain reaction (RT-PCR) reagents were from Applied Biosystems by Thermo Fisher Scientific, Carlsbad, CA, USA. HDGF siRNA, sc-45878, was obtained from Santa Cruz Biotechnology, Dallas, TX, USA. Control siRNA (sc-37007, Santa Cruz, Dallas, TX, USA). Primary antibodies used: (a) mouse monoclonal: Anti-β-Actin Antibody (C4): sc-47778 (Santa Cruz Biotechnology, Dallas, TX, USA); (b) rabbit monoclonal and polyclonal: Anti-HDGF (E3P7K): 42105 (Cell Signaling Technology, Danvers, MA, USA).

### 4.2. In Vitro Studies

We performed most of the in vitro experiments on three major human cell lines to demonstrate the effects of apoptotic hepatocytes/lymphocytes on HSC. Given the paucity of primary human cells and large numbers of cells necessary for the generation of ABs, Huh7.5 cells with attenuated innate immunity overexpressing CYP2E1 (designated as RLW cells) were used instead of PHH, while Jurkat cells were used instead of human lymphocytes, and LX2 cells were used instead of HSC. Although RLW cells are considered as hepatocyte-like, they are alcohol dehydrogenase (ADH) deficient. Consequently, RLW cells inadequately metabolize alcohol and do not make enough acetaldehyde to mimic the natural ethanol metabolism. Thus, RLW cells were treated with a culture medium containing an exogenous source of acetaldehyde. This exogenous source of continuously released acetaldehyde, called the acetaldehyde generating system (AGS), consists of 0.02 EU of yeast and ADH metabolizes 50 mM of ethanol in the presence of 22 mM nicotinamide adenine dinucleotide (NAD) as a co-factor [10]. HIV used in this study to infect RLW cells was propagated and purified from primary human macrophages at the University of Nebraska Medical Center, Omaha, NE, USA [10]. 

Dr. Laura Schrum (Carolinas Healthcare System, Charlotte, NC, USA) generously provided the LX2 cell line used for this study, and the Jurkat cells were provided by Dr. Wanfen Xiong at the University of Nebraska Medical Center, Omaha, NE, USA. To generate both hepatocyte and lymphocyte ABs, RLW and Jurkat cells were pre-treated or not with AGS for 24 h, infected or not with HIV 1_ADA_ (MOI = 0.1) overnight, and then exposed or not to AGS for four days. ABs-containing medium was collected in a 50 mL conical tube after four days and processed immediately for ABs. Isolation of ABs was performed by the differential centrifugation method previously described [11]. First, the cell debris in the ABs-containing medium was separated by 300× *g* centrifugation at 4 °C for 10 min. The cell debris-free medium was then collected into a new 50 mL conical tube for another centrifugation. Here, the centrifugation was carried out at 3000× *g* for 20 min to isolate ABs. The purity, concentration, and size of the ABs have been characterized as we previously published [11]. The numbers of isolated ABs from Jurkat and RLW cells were determined by nanoparticle tracking analysis (NTA). Then, equal amount of RLW or Jurkat cell ABs were introduced to LX2 cells at the ratio of 3:1 and incubated for 2 h. Profibrotic gene measurements were carried out by RT- PCR in LX2 cells after 2 h of ABs exposure. 

### 4.3. siRNA Transfection of LX2 Cells

Inhibition of HDGF in RLW cells was achieved by HDGF siRNA transfection following the manufacturer’s instructions. Four hours before the siRNA application, FBS-rich medium (DMEM, 2% FBS, and 1% Penicillin streptomycin) was replaced with FBS-deficient media (DMEM and 1% Penicillin streptomycin) in the LX2 culture system. Then the cells were incubated with siRNA in the transfection media (sc-36868, Santa Cruz, Dallas, TX, USA) for 5–6 h. The efficiency of siRNA transfection was evaluated using scrambled siRNA-FITC Conjugate-A (sc-36869, Santa Cruz, Dallas, TX, USA), and approximately 93.75% of siRNA FITC-positive cells was detected by immunostaining (Figure 9). After successful transfection of HDGF siRNA, the same cells were pretreated with AGS for 24 h, then HIV overnight and AGS for 48 h. HDGF-siRNA-transfected cells could not be incubated for more than 48 h due to the chance of reactivating the silenced HDGF gene. Control siRNA (sc-37007, Santa Cruz, Dallas, TX, USA) was included in the experiment as a negative control.

### 4.4. Sample Preparation for Mass Spectrometry

The protein concentration was estimated in each sample using a BCA Protein Assay Kit (Pierce). An amount of 100 μg of protein from each sample was diluted to 100 μL volume with 100 mM of ammonium bicarbonate (ambic). Proteins were reduced with 5 μL of 200 mM tris(2-carboxyethyl) phosphine (TCEP) (1 h incubation, 55 °C) and alkylated with 5 μL of 375 mM iodoacetamide (IAA) (30 min incubation in the dark, room temperature). The reduced and alkylated proteins were purified with acetone precipitation at −20 °C overnight. The next day, protein precipitates were collected by centrifugation at 8000× *g* for 10 min at 4 °C, and pellets were briefly air-dried and resuspended in 100 μL of 50 mM ambic. The protein digestion was carried out using 2.5 μg of trypsin per sample (16 h incubation, 37 °C). The next day, samples were dried out using a speed vacuum and then desalted with C18 spin columns (Pierce). Clean peptides were dried out again with speed vacuum and resuspended in 0.1% formic acid, and next analyzed using a high-resolution mass spectrometry nano-LC-MS/MS Tribrid system, Orbitrap Fusion™ Lumos™ coupled with UltiMate 3000 HPLC system (Thermo Scientific). 

### 4.5. LC-MS/MS 

An amount of 1.5 μg of each sample was loaded onto trap column Acclaim PepMap 100 (75 μm × 2 cm C18 LC Columns, Thermo Fisher Scientific) at a flow rate of 4 μL/min, and next separated with a Thermo RSLC Ultimate 3000 (Thermo Fisher Scientific) on a Thermo Easy-Spray PepMap RSLC C18 column (75 μm × 50 cm C-18 2 μm, Thermo Fisher Scientific) at a flow rate 0.3 μL/min and 50 °C, with a step gradient of 9%–25% solvent B (0.1% FA in 80% acetonitrile) from 10–15 min and 25%–40% solvent B from 15–40 min, with a 70 min total run time. The MS scan was conducted using a detector: Orbitrap resolution 120,000; scan range 350–1800 m/z; RF lens 30%; AGC target 4.0 × 10^5^; maximum injection time 100 ms. The most intense ions with a charge state of 2–6 ions isolated in 3 s cycles were selected in the MS scan for further fragmentation. MS2 scan parameters set activation HCD with 35% normalized collision energy, detected at a mass resolution of 30,000. The AGC target for MS/MS was set at 5.0 e4 and ion filling time set to 60 ms.

### 4.6. Data Analysis and Bioinformatics Analysis

Protein identification was performed by searching MS/MS data against the UniProt datABsase (selected for Human+HIV) in Proteome Discoverer (Thermo Fisher Sci, vs. 2.5.), assuming the digestion enzyme trypsin. The parameters for Sequest HT were set as follows: enzyme: trypsin, max missed cleavage: 2, precursor mass tolerance: 10 ppm, peptide tolerance: ±0.02 Da, fixed modifications: carbamidomethyl (C); dynamic modifications: oxidation (M), acetyl (N-term). The parameters for the precursor ion quantifier were set as follows: peptides to use unique + razor, precursor abundance based on intensity; normalization mode: total peptide amount; scaling mode: on all averages. Student’s *t*-test was used for statistical analysis. All data are expressed as the mean ± SE, and a *p*-value < 0.05 was considered to indicate statistical significance. The protein expression data are presented in a heatmap and volcano plot. The expression between groups was defined as significantly upregulated or downregulated based on a fold change (FC)  ≥ 2 and *p*-value < 0.05. 

### 4.7. RNA Isolation and RT-PCR

RNAs encoding *HIV gag*, *TIMP1*, *TGFβ*, *COL1A1*, and *MMP2* were measured by RT-PCR as described before [10]. Total RNA was isolated from cells by the Trizol method. This was carried out using the two-step procedure involving the transcription of at least 200 ng of RNA to cDNA with a high-capacity reverse transcription kit (Applied Biosystems, ThermoFisher Scientific, Waltham, MA, USA), and the resulting cDNA transcript was amplified using TaqMan Universal Master Mix II with fluorescent-labeled primers (TaqMan gene expression systems) by a Model 7500 qRT-PCR thermal cycler. The threshold cycle (Ct) of the RNA transcript was subtracted from the Ct of the reference, GAPDH, and then used to calculate the relative quantity of each RNA transcript. 

### 4.8. Immunoblotting/Western Blotting 

Immunoblotting was performed as described previously [11]; blots were developed using 27444 Bio-Rad Imaging System Chemidoc Touch Imaging System, and protein band densities were quantified using the NIH Image J software program. Equal (20 µg) amounts of protein were loaded in each lane. β-actin was used as the loading control for normalization. 

### 4.9. In Vivo Studies

#### 4.9.1. Humanized Mice

We used NOD.Cg-Prkdcscid Il2rgtm1Wjl/SzJ (NSG) mice transplanted with human umbilical cord-derived hematopoietic stem cells CD34+. We call them humanized animals as they carry the human immune system. NOD/scid-IL-2Rγcnull mice were obtained from Jackson Laboratories (Bar Harbor, ME, USA, stock #05557). They were bred under specific pathogen-free conditions in accordance with ethical guidelines for the care of laboratory animals at the UNMC as set forth by the National Institutes of Health. Mice were transplanted at birth with human hematopoietic stem cells isolated from umbilical cord blood, as described [38]. Briefly, we irradiated newborn pups on days 1–2 at 1 Gy and transplanted 10^5^ CD34+ cells intrahepatically. We evaluated human cells’ engraftment in 16–20 week-old mice by staining peripheral blood for human pan-CD45, CD3, CD4, and CD8 markers as four-color combinations. Antibodies and isotype controls were obtained from BD PharMingen (San Diego, CA, USA), and staining was analyzed with a FACSDiva (BD Immunocytometry Systems, Mountain View, CA, USA). Results were expressed as percentages of the total number of gated lymphocytes. The gating strategy was human CD45→CD3→CD4/CD8.

On day 24, after initiating ethanol in drinking water as described [39], the control and alcoholic mice were infected with HIV-1_ADA_ intraperitoneally at 10^4^ TCID50 per mouse. The levels of viral RNA copies/mL were analyzed by an automated COBAS Amplicor System (Roche Molecular Diagnostics, Basel, Switzerland). For assay use, mouse plasma samples (20 μL each) were diluted to 350 μL with a normal human serum which increased the detection limit to 350 viral RNA copies/mL. HIV-1 infection was confirmed by virologic and histological examination in 8 animals. One reconstituted animal not exposed to HIV-1 served as the control. Animals were euthanized at 5 weeks after infection. No mortalities were induced by HIV-1 infection and alcohol consumption. The levels of ALT and AST were analyzed by the clinical laboratory standard tests. 

#### 4.9.2. Immunohistochemistry 

Livers were removed immediately after euthanasia and processed. Tissue samples were fixed with 4% paraformaldehyde overnight and embedded in paraffin. Five-micron-thick sections were stained with hematoxylin and eosin (H&E), and mouse monoclonal antibodies for HLA-DQ/DP/DR (clone CR3/43, 1:100), HIV-1 p24 (clone Kal-1, 1:10). Mouse monoclonal antibodies to alpha-smooth muscle actin (α-Sma) were obtained from ABscam (clone 1A4, 1:500 dilution, Cambridge, UK). Polymer-based horseradish peroxidase-conjugated anti-mouse systems were used as secondary detection reagents and were developed with 3,3′-diaminobenzidine (DABS). All paraffin-embedded sections were counterstained with Mayer’s hematoxylin. Bright-field images were obtained with a Leica DM1000 LED under an original magnification of 100×. The staining for connective tissue was carried out with a Picro Sirius Red stain kit (Abcam, 150681) according to manufacturer instructions. 

#### 4.9.3. Statistical Analyses

Data were analyzed using GraphPad Prism v7.03 software (GraphPad, La Jolla, CA, USA). Data from at least three duplicate independent experiments were expressed as mean ± SEM. Comparisons among multiple groups were performed by one-way ANOVA, using a Tukey posthoc test. For comparisons between the two groups, we used Student’s *t*-test. A *p*-value of 0.05 or less was considered significant.

## 5. Conclusions

We conclude that the profibrotic activation of HSC can be achieved by the engulfment of ABs of hepatocyte origin generated from hepatocytes, but not lymphocytes exposed to HIV and AGS. These ABs contain not only HIV proteins, but also hepato-specific factors, such as HDGF. 

## Figures and Tables

**Figure 1 ijms-24-05346-f001:**
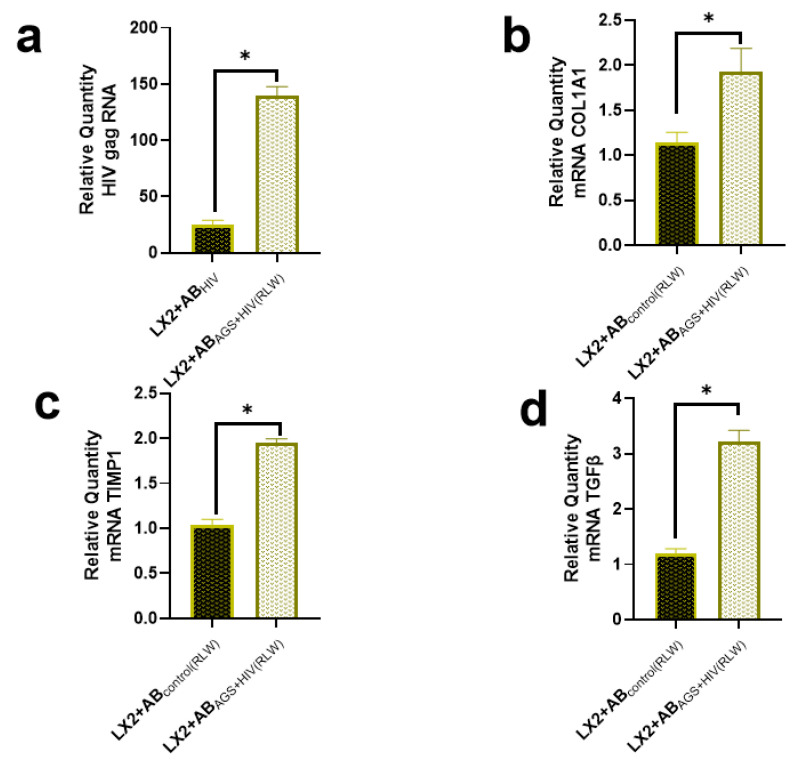
ABs derived from AGS-treated and HIV-infected RLW cells enhance HIV expression in LX2 cells as well as induce profibrotic changes: (**a**) RT-PCR analysis of HIV gag RNA expression in LX2 cells after internalization of AB_AGS+HIV_. RT-PCR analysis of (**b**) *COL1A1* mRNA, (**c**) *TIMP1* mRNA, and (**d**) *TGFβ2* mRNA of LX2 cells exposed to RLW AB_AGS+HIV_ for 2 h. Bars with * are significantly different from each other at *p* ≤ 0.05. Data were generated from 3 independent experiments.

**Figure 2 ijms-24-05346-f002:**
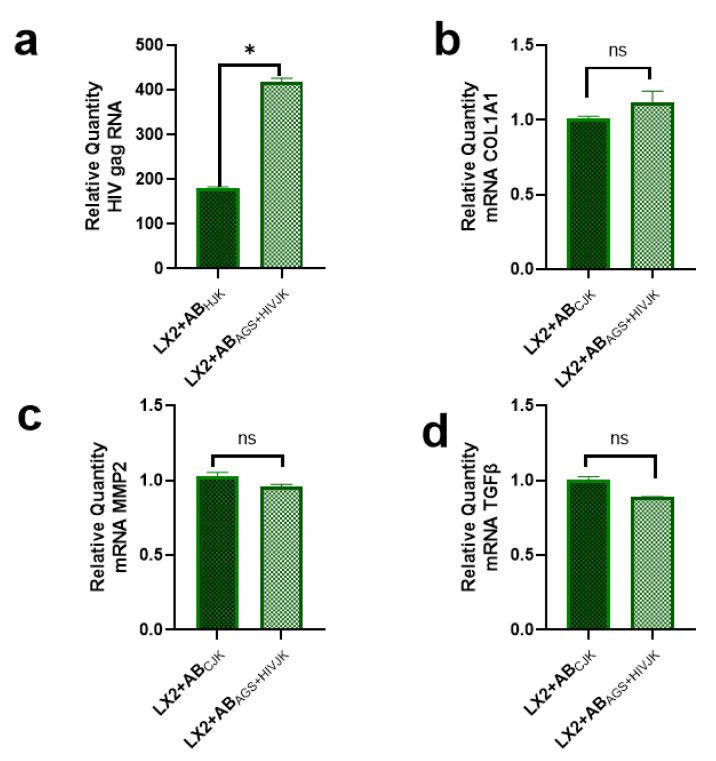
ABs derived from acetaldehyde and HIV-infected Jurkat cells enhance HIV expression but do not trigger profibrotic activation in LX2-cells: RT-PCR quantification of (**a**) *HIV gag* RNA, (**b**) *COL1A1* mRNA, (**c**) *MMP2* mRNA, and (**d**) *TGFβ1* mRNA. Data are from 3 independent experiments presented as mean ± SEM. Bars with * are significantly different from each other and bars marked ns are not statistically different at *p* ≤ 0.05.

**Figure 3 ijms-24-05346-f003:**
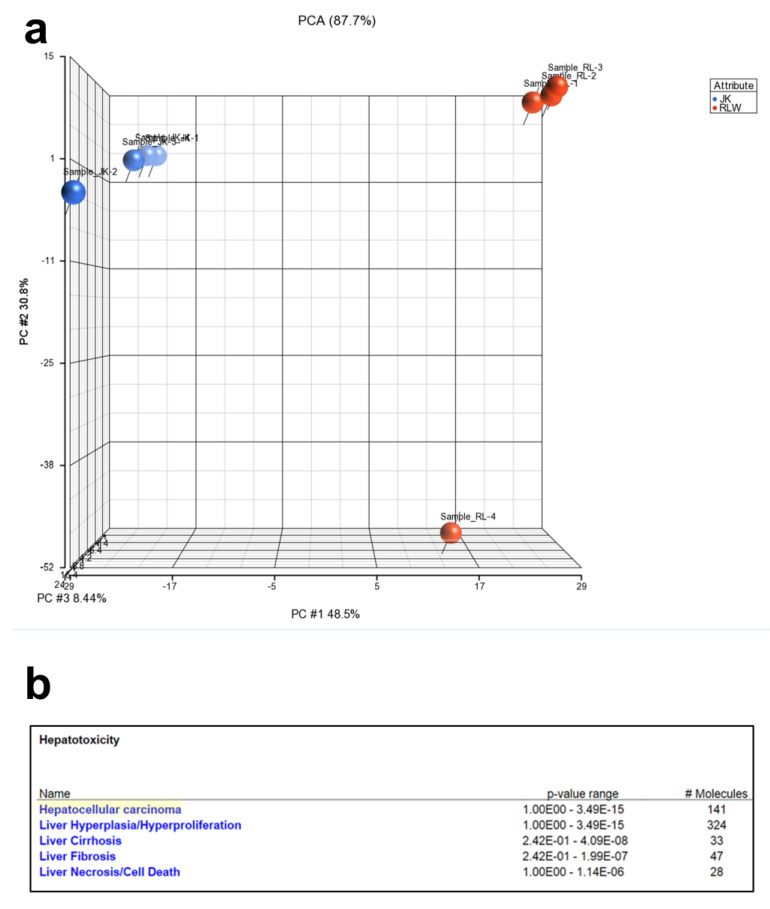
Protein profile of lymphocyte ABs versus hepatocyte ABs analyzed by mass spectrometry. (**a**) Principal component analysis of all analyzed proteins showed that hepatocyte ABs are distinct from lymphocyte ABs. Orange pins represent RLW-derived ABs, and blue pins represent Jurkat cell-derived Abs. (**b**) Ingenuity pathway analysis identified proteins, which are significantly associated with hepatoxicity (*p* < 0.05). # indicate number. Data were generated from 4 independent experiments.

**Figure 4 ijms-24-05346-f004:**
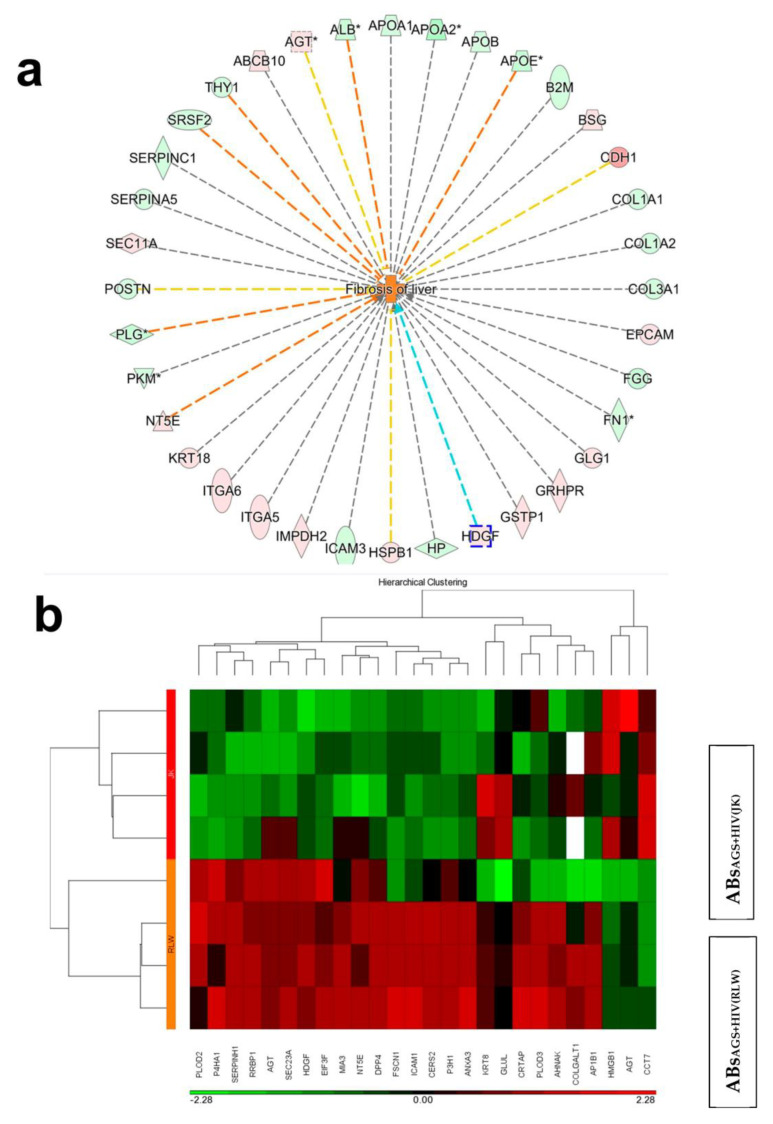
Characterization of proteins associated with liver fibrosis in hepatocyte vs. lymphocyte ABs by mass spectrometry (**a**) ingenuity pathway analysis showing the top 36 proteins associated with liver fibrosis. The green color represents downregulated proteins in AB_AGS+HIV(RLW)_ vs. AB_AGS+HIVJK_ and the pink color represents upregulated proteins of AB_AGS+HIV(RLW)_ vs. AB_AGS+HIVJK_. Proteins with asterisk (*) indicates that several identifiers in the data link to a single gene in the Global Molecular Network (**b**) Heatmaps of top 26 proteins of hepatocyte ABs vs. lymphocyte ABs. The red color represented upregulated proteins of AB_AGS+HIV(RLW)_ vs. AB_AGS+HIVJK_; the green color represented downregulated protein of AB_AGS+HIV(RLW)_ vs. AB_AGS+HIVJK_; black color represented an equal amount of proteins in AB_AGS+HIV(RLW)_ and AB_AGS+HIVJK_. Data significantly different from each other (*p* ≤ 0.05) are from 4 independent experiments.

**Figure 5 ijms-24-05346-f005:**
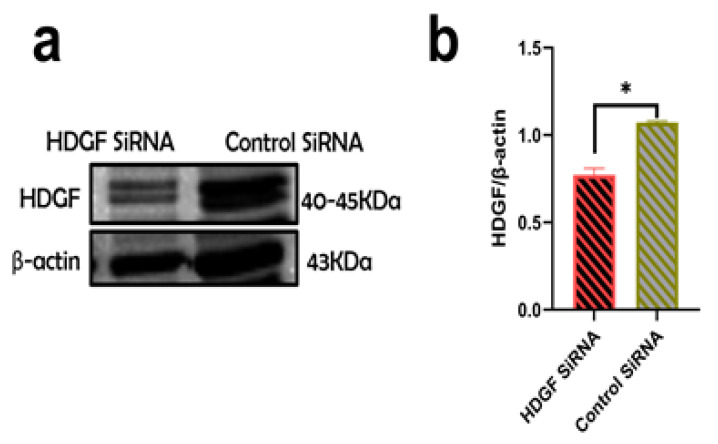
Efficiency of HDGF suppression by siRNA: (**a**). Transfection of RLW cells with HDGF siRNA. HDGF was measured in cell lysates of HDGF- or control-siRNA-transfected RLW cells by immunoblot analysis (**b**) Quantification of immunoreactive bands by NIH ImageJ. Data are from 3 independent experiments presented as mean ± SEM. Bars with * are significantly different from each other (*p* ≤ 0.05).

**Figure 6 ijms-24-05346-f006:**
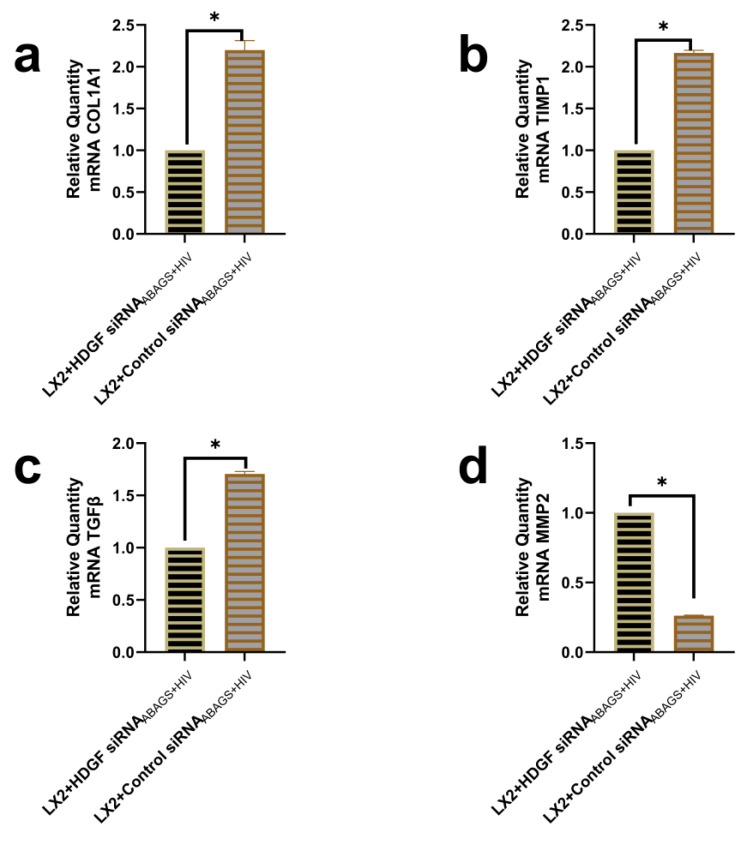
Silencing HDGF in RLW ABs attenuated profibrotic genes in LX2 cells. RT-PCR analysis of (**a**) COL1A1 mRNA, (**b**) *TIMP1* mRNA, and (**c**) *TGFβ* mRNA. (**d**) *MMP2* mRNA. Data are from 3 independent duplicate experiments presented as mean ± SEM. Bars with * are significantly different from each other (*p* ≤ 0.05).

**Figure 7 ijms-24-05346-f007:**
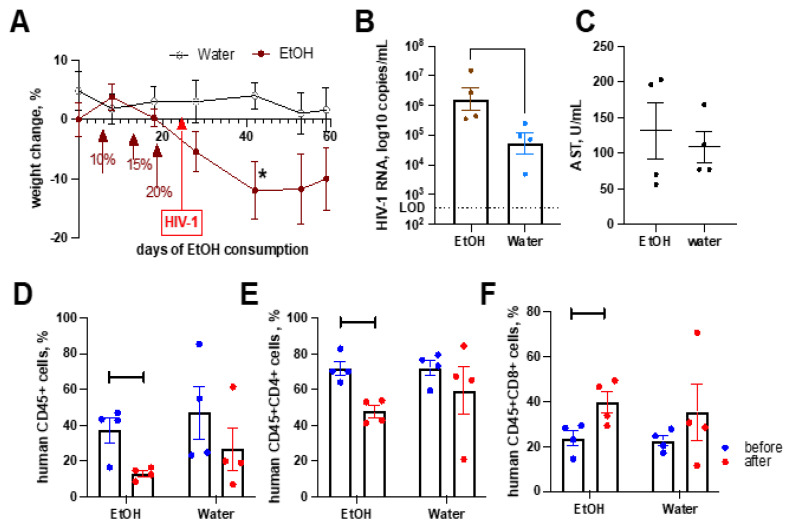
Alcohol consumption worsens HIV-1-induced immunopathology but does not change liver enzyme activity. (**A**) Humanized animals exposed to increased concentrations of alcohol in water showed a reduction in body weight compared to control mice kept on regular water. (**B**) Alcohol-consuming mice had a higher viral load in peripheral blood. (**C**) The activity of AST remained in the normal range, and ALT was the same at 49 U/mL in all animals. (**D**–**F**) Alcohol accelerated the development of immunopathology. The number of human cells in the peripheral blood of alcoholic mice dropped due to HIV-1 infection (**D**). The depletion of CD4+ cells (**E**) and increased CD8+ cell count (**F**) also was significant in ethanol-consuming animals. (**B**–**F**) mean and SEM, and individual numbers are shown. Statistical significance was determined using GraphPad Prism Version 9.4.1 by Mann–Whitney tests, and only significant changes are indicated: *—*p* < 0.05, Each group has 4 mice.

**Figure 8 ijms-24-05346-f008:**
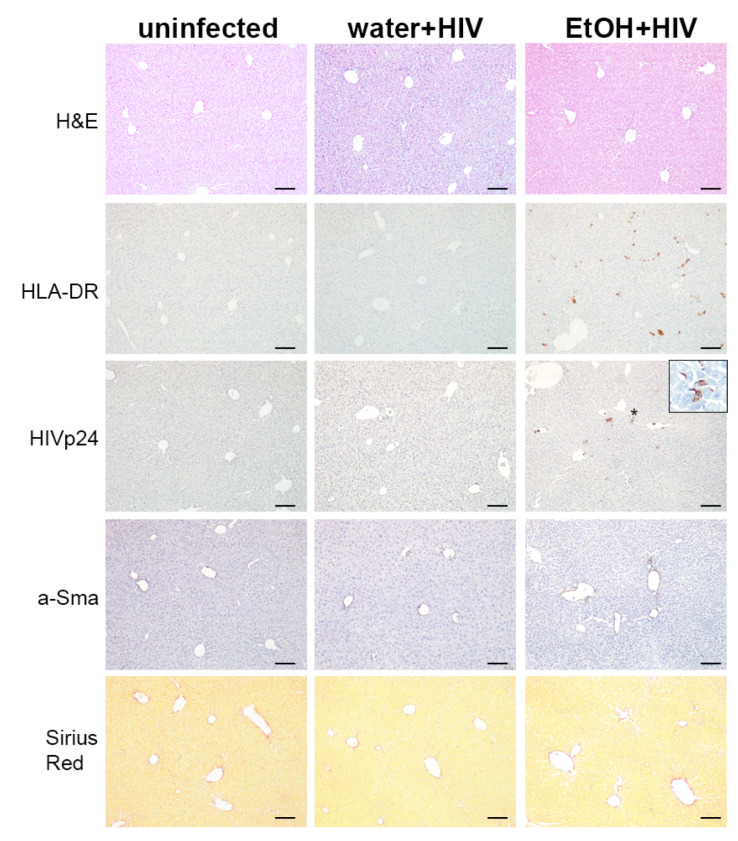
Liver pathomorphology in HIV-1 infected animals consuming alcohol in water. Humanized mice were fed 20% *w*/*v* alcohol in drinking water and infected with HIV-1 for 5 weeks. Liver tissues of uninfected and HIV-1-infected water- or alcohol-drinking mice were fixed, paraffin-embedded, and 5 μm sections were stained for H&E, human HLA-DR, HIVp24, a-SMA, and Sirius Red, as indicated. Human cells positive for HLA-DR and HIVp24 staining were observed in the parenchyma, and some infected cells (star) were apoptotic, as shown in a magnified view (inset) of the boxed area. There were no differences in stellate cell activation or collagen deposition. Scale bars: 100 μm. Each mouse group had 4 mice.

**Figure 9 ijms-24-05346-f009:**
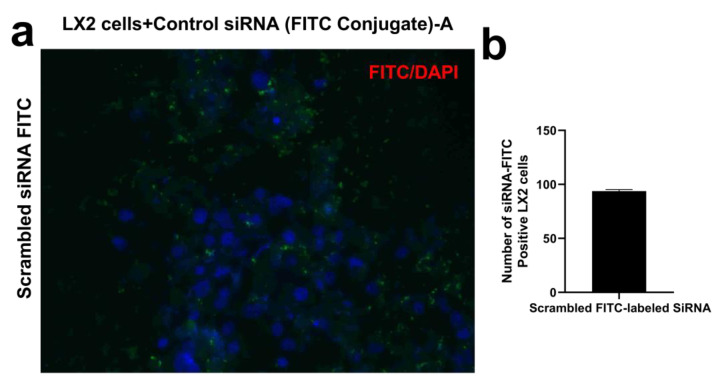
Transfection of LX2 cells with siRNA. (**a**)The transfection efficiency of LX2 cells was measured by quantifying the siRNA-scrambled FITC (green)-positive cells. Cells were visualized by a 20× Keyence microscope. The picture used is representative data from one out of three independent experiments. (**b**) Transfection efficiency quantification.

## Data Availability

The datasets presented in this study are available from the corresponding author upon reasonable request.

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
