# Peer review of "Hepatocyte-Specific Triggering of Hepatic Stellate Cell Profibrotic Activation by Apoptotic Bodies: The Role of Hepatoma-Derived Growth Factor, HIV, and Ethanol"

_ijms, 2023, doi:10.3390/ijms24065346_

Round 1

Reviewer 1 Report

The authors analyzed lymphocyte-derived ABs to trigger HSC profibrotic activation in comparison to hepatocyte-derived ABs. ABs cargo was analysed by proteomics.

The authors demonstrate that ABs generated from RLW, but not from Jurkat cells activated fibrogenic genes in HSC and state that hepatocyte-specific proteins like Hepatocyte Derived Growth Factor attenuates pro-fibrotic activation of HSC.

The authors aimed to demonstrate the effect in mice humanized with only immune cells but not with human hepatocytes.

It was concluded that ABs of hepatocyte origin promote HSC activation and liver fibrosis development.

Comments:

-error bars for controls are missing, e.g. in Fig. 1 b-d and Fig. 2 b-d

-the number of technical and biological replications per group should be presented in each figure legend (for in vitro and in vivo experiments)

-all proteomics data should be published in parallel in an appropriate database

-the positive control for hepatic fibrosis in the in vivo experiment is missing, i.e. HIV-infection and ethanol diet in mice with transplanted human hepatocytes

-the authors should discuss their results intensively in the context of the current literature, e.g. Akil et al. DOI:10.1038/s41598-018-37071-y

Author Response

Response to reviewers

We thank the reviewers for their thoughtful comments and would like to respond to the identified issues. The changes in the text are highlighted in yellow.

 Reviewer 1:

  1. Error bars for controls are missing, e.g. in Fig. 1 b-d and Fig. 2 b-d

Response: Error bars were added to the indicated figures

  1. All proteomics data should be published in parallel in an appropriate database

Response: The raw proteomics data have been uploaded in MassIVE repository. Dataset MSV000091328 is currently private. It has been deposited to MassIVE, but has not yet been publicly released.

Submission details:
Project Name:  Hepatocyte-Specific Triggering of Hepatic Stellate Cell Profibrotic Activation by Apoptotic Bodies: The Role Of Hepatoma Derived Growth Factor, HIV, and Ethanol  
Pride accession: PXD040290

MassIVE accession ID- MSV000091328
Reviewer account details:

Username: MSV000091328_reviewer

Password: mspcf

  1. The number of technical and biological replications per group should be presented in each figure legend (for in vitro and in vivo experiments)

Response: For in vitro studies, we used three independent duplicate experiments; for proteomics analysis, we used four independent experiments. For in vivo studies, each group contained of four humanized mice. Now, this is included in figure legends.

  1. The positive control for hepatic fibrosis in the in vivo experiment is missing, i.e. HIV infection and ethanol diet in mice with transplanted human hepatocytes

Response. HIV is a human virus, which enters only human cells and replicates in human cells.  Previously, we have shown that hepatocytes do not productively replicate HIV, but exposure to ethanol stabilizes HIV in these cells due to the suppression of protein-degrading intracellular systems. Thus, the transplantation of human hepatocytes without the human immune system does not provide a long-term stable source of HIV infection, and hepatocytes will not be exposed to high levels of HIV expressed in immune cells, while the activation of liver fibrosis requires continues exposure of liver cells to HIV and ethanol metabolites. This can be established in mice transplanted with both human hepatocytes and human immune cells. Unfortunately, so far, we were not able to establish the syngeneic system where mice were humanized with both human hepatocytes and a human immune system, HIV-infected, and fed an ethanol diet to induce liver fibrosis that can be used as a positive control. However, when immunodeficient mice were ethanol- fed and injected with HIV-containing human hepatocyte apoptotic bodies (ABs), we observed some pro-fibrotic changes   (doi.org/10.3390/ biology11071059).  In the current study, when we tried to humanize mice only with immune cells, infection with HIV and exposure to alcohol diets provided no evidence of liver fibrosis development, thereby indicating that in addition to HIV-infected immune cells and liver macrophages, this process requires the presence of “extra elements”, which might be human hepatocytes. In fact, as we mentioned in the paper, based on clinical and epidemiological data, liver fibrosis development was identified in HIV-infected patients and its risk is significantly increased in alcohol abusers.

  1. The authors should discuss their results intensively in the context of the current literature, e.g. Akil et al. DOI:10.1038/s41598-018-37071-y

Response: We thank the reviewer for providing a reference to this excellent paper. Interestingly, in our previous studies, we observed the formation of ABs from HIV-infected hepatocytes exposed to second hits (like alcohol or HCV co-infection, doi: 10.3390/biom9120851 and PMID: 29679566). HSC activation by engulfment of these ABs was further confirmed in detail by our other studies (doi: 10.3390/biom11101497; doi.org/10.3390/ biology11071059). For liver fibrosis, there is no doubt about the importance of the crosstalk between HSC, macrophages, and hepatocytes via cytokines and EVs in the context of alcohol exposure (doi.org/10.3389/fphys.2022.831004). Furthermore, we cannot exclude that ABs engulfment plays a role in M-L-H interactions demonstrated in DOI:10.1038/s41598-018-37071-y in activation the mechanisms suggested in our current manuscript. However, in this study, we investigated the direct HSC induction by HIV-acetaldehyde -exposed hepatocyte ABs without the involvement of macrophages. Importantly, in doi: 10.3390/biom11101497, in a separate experiment, we have shown that internalization of hepatocyte HIV+ AB by macrophages promotes inflammasome induction in these cells, further pertaining to pro-fibrotic HSC activation. Anyway, the involvement of hepatocytes in the activation of both types of non-parenchymal cells is critical for fibrosis development. Now, we included these explanations in the Discussion.

Reviewer 2 Report

The risk of liver fibrosis development is potentiated by alcohol abuse and HIV infection. In the previous studies, the authors reported that hepatocytes exposed to HIV and acetaldehyde undergo significant apoptosis, and the engulfment of apoptotic bodies (ABs) by hepatic stellate cells (HSC) potentiates their pro-fibrotic activation. As ABs can be generated from liver-infiltrating immune cells, they explored whether lymphocyte-derived ABs trigger HSC profibrotic activation as strongly as hepatocyte-derived ABs. They found that ABs generated from Huh7.5- CYP2E1 (RLW) cells, but not from Jurkat cells activated fibrogenic genes in HSC, which was driven by the expression of hepatocyte-specific proteins, such as HDGF in ABs cargo. HDGF Knockdown attenuates pro-fibrotic activation of HSC. In mice humanized with only immune cells but not human hepatocytes, infected with HIV, and fed ethanol, liver fibrosis was not observed. They concluded that ABs of hepatocyte origin promoted HSC activation and liver fibrosis development. The description is coherent, except the description of NSG mice and humanized model are quite confusing. Besides, there are many concerns that still need to be addressed.

1.     “To explore whether lymphocyte-derived ABs trigger HSC profibrotic activation as strongly as hepatocyte-derived ABs”, using only Jurkat cells-derived ABs is inadequate to get a clear conclusion if the result is Jurkat cells specific or other lymphocytes should have a similar effect as well.

2.     Kupffer cells inside the liver are the critical immune cells for HSC activation, and there have been many reports of the ABs from macrophages play key roles in many disease development. It will be interesting to test the effect of macrophage ABs on HSC activation.

3.     For reproductivity, another siRNA should be included for the HDGF knockdown assays.

4. In Line 109, the same number of ABs and the same ratio were used for the testing. Did the authors ever check if the size and size distribution of ABs from hepatocytes and Jurkat cells were similar or not?

5.     As knockdown of HDGF represses the profibrotic gene expression in LX2 cells, it will be interesting to test if ectopic overexpression of HDGF in Jurkat or other lymphocytes will affect profibrotic gene expression in LX2 cells through their new ABs.

6. In Lines 207, 234, 236, if it is not the regular “NSG mice” used in the experiments, please rephrase to exactly reflect the mice tested. What is the difference between the “NSG mice” and the “Humanized animals” (Line 221, 232)? What kind of mice were exactly used?

Author Response

Response to reviewers

We thank the reviewers for their thoughtful comments and would like to respond to the identified issues. The changes in the text are highlighted in yellow.

Reviewer 2:

  1. “To explore whether lymphocyte-derived ABs trigger HSC profibrotic activation as strongly as hepatocyte-derived ABs”, using only Jurkat cells-derived ABs is inadequate to get a clear conclusion if the result is Jurkat cells specific or other lymphocytes should have a similar effect as well.

Response: In our previous (already published) study (doi: 10.3390/biom11101497, Fig. 8 D), we have shown that when HSC engulfed human apoptotic primary lymphocytes infected with HIV, we observed even decrease instead of an increase in expression of pro-fibrotic markers. However, at that point, we did not expand our studies to characterize the hepatospecific markers that induce HSC activation. For proteomics studies on isolated ABs, we needed more cells than we were able to establish for primary lymphocytes. That is why while this phenomenon was originally demonstrated using primary human lymphocytes, it was further confirmed/studied on the cell line (Jurkat cells) infected with HIV.

  1. Kupffer cells inside the liver are the critical immune cells for HSC activation. Many reports of the ABs from macrophages play key roles in many diseases development. It will be interesting to test the effect of macrophage ABs on HSC activation.

Response: While the crosstalk between hepatocytes, macrophages, and HSC is extremely important for liver fibrosis development as demonstrated in HIV-HCV co-infection model (DOI:10.1038/s41598-018-37071-y), in our study, we concentrated on the interaction between hepatocytes and HSC in the context of HIV-infection and exposure to ethanol metabolites. However, we cannot exclude the potentiating role of macrophages in this process. In fact, in our already published studies (doi: 10.3390/biom11101497, Fig, 8A), we have shown that the engulfment of HIV-infected apoptotic hepatocytes by human macrophages induced inflammasome activation, which, theoretically, may further potentiate pro-fibrotic HSC activation. This will be the focus of our future studies on AB-initiated crosstalk in HIV/alcohol-induced liver fibrosis development.

  1. For reproductivity, another siRNA should be included for the HDGF knockdown assays.

Response: For our studies, we used siRNAs probes from Santa Cruz, which was a mixture of several siRNAs. The siRNA we used is a pool of 3 target-specific oiligos as per the manufacturer’s datasheet (Catalogue # SC-45878). The data has been reproduced with a statistically very significant knockdown as shown by western blot (Fig-5 and Fig-9, with the transfection efficiency of 93.75%).

  1. In Line 109, the same number of ABs and the same ratio were used for the testing. Did the authors ever check if the size and size distribution of ABs from hepatocytes and Jurkat cells were similar or not?

Response: The size of ABs in hepatocyte ABs was characterized in detail in our published studies (doi.org/10.3390/ biology11071059).  However, the sizes of Jurkat AB were characterized by another author and expected to be smaller than hepatocyte AB since Jurkat cells are at least two folds smaller than hepatocytes, and the size of ABs depends on cell size (doi: 10.1080/20013078.2019.1608786). Despite of size, these Jurkat cell-generated ABs contained no HDGF and were successfully engulfed by HSC.

  1. As knockdown of HDGF represses the profibrotic gene expression in LX2 cells, it will be interesting to test if ectopic overexpression of HDGF in Jurkat or other lymphocytes will affect profibrotic gene expression in LX2 cells through their new ABs.

Response: RLW cells express quite high levels of HDGF. However, in this short period for the manuscript revision, we were unable to obtain and expand HDGF plasmid for overexpression. While this approach might be useful, it is unlikely that this overexpression will make a huge difference in the interpretation of our results.  However, this is an excellent suggestion for our future experiments, and we will validate the role of not only HDGF, but also other genes/proteins expressed in hepatocytes, not in lymphocytes, which are related to liver fibrosis induction.   

  1. In Lines 207, 234, 236, if it is not the regular “NSG mice” used in the experiments, please rephrase to exactly reflect the mice tested. What is the difference between the “NSG mice” and the “Humanized animals” (Line 221, 232)? What kind of mice were exactly used?

Response: We used NOD.Cg-Prkdcscid Il2rgtm1Wjl/SzJ (NSG) mice transplanted with human umbilical cord derived hematopoietic stem cells CD34+. We call them Humanized animals as they carry the human immune system. This explanation is in the Material and Method section.

Round 2

Reviewer 1 Report

The revised version has improved.

Nevertheless, the authors state in their abstract ´We conclude that ABs of hepatocyte origin promote HSC activation and liver fibrosis development.´, but ´promotion of liver fibrosis development by ABs of hepatocytes´ is not demonstrated.

I aimed to adress this point with the fourth comment of my first revision. Maybe, this comment was not clear enough. I thank the authors for their intructive answer but data, which might strengthen the fact that ´liver fibrosis development is promoted by ABs of hepatocytes´ are still missing in the revised version of this manuscript.

Even the citation which was provided with the answer to this point is not available: doi.org/10.3390/ biology11071059 ´DOI cannot be found in the DOI System.´

Development of hepatic fibrosis can be demonstrated by the quantification of the increase of fibrillar collagen, i.e. by the quantification of hydroxyproline in liver lysates. The authors clearly demonstrated that ABs of hepatocyte origin but not lymphocyte-derived ABs promote HSC activation. They even demonstrated that ABs of hepatocyte origin increased COL1A1 mRNA in LX2.

This point should be revised accordingly.

e.g. in the abstract may be stated: ´We conclude that ABs of hepatocyte origin but not lymphocyte-derived ABs promote HSC activation and increase levels of COL1A1 mRNA.´.

Author Response

Response to reviewers (round 2):

We thank the reviewers for the time they spent reviewing this manuscript and for their helpful suggestions. We would like to provide some explanations to Reviewer 1 regarding his comments. The changes to the paper are highlighted in yellow.

  1. ´promotion of liver fibrosis development by ABs of hepatocytes´ is not demonstrated. Even the citation which was provided with the answer to this point is not available: doi.org/10.3390/ biology11071059 ´DOI cannot be found in the DOI System.´

Response: We apologize for the glitch with DOI system. The full citation of this article is:

Moses New-Aaron, Raghubendra Singh Dagur, Siva Sankar Koganti, Murali Ganesan, Weimin Wang, Edward Makarov, Mojisola Ogunnaike, Kusum K. Kharbanda, Larisa Y. Poluektova and Natalia A. Osna Alcohol and HIV-Derived Hepatocyte Apoptotic Bodies Induce Hepatic Stellate Cell Activation. Biology 2022, 11, 1059., PMID: 36101437

In this article, we demonstrated in vivo model where mice were fed control or ethanol diets and multiply injected with ABs generated from human HIV- infected or non-infected RLW. The combination of HIV+ hepatocyte AB and ethanol feeding induced activation of Col1A1 and TGFβ genes and initiated positive Sirius Red staining. This was interpreted as a progression to liver fibrosis induced by the combination of HIV+ hepatocyte ABs and ethanol exposure.  

  1. g. in the abstract may be stated: ´We conclude that ABs of hepatocyte origin but not lymphocyte-derived ABs promote HSC activation and increase levels of COL1A1 mRNA.

Response: We fully understand the concerns of the reviewer since liver fibrosis development is a long process that takes years in alcohol-abusing PLWH. As per the reviewer’s suggestion, we removed “fibrosis development” from the abstract rephrasing the conclusion as follows: We conclude that HIV+ ABs of hepatocyte origin promote HSC activation, which potentially may lead to liver fibrosis progression. 

Reviewer 2 Report

Thank you for addressing all the concerns raised in the initial review. There is no further comment.

Author Response

Thank you to reviewer for accepting this paper.